# Challenges in Combining Immunotherapy with Radiotherapy in Recurrent/Metastatic Head and Neck Cancer

**DOI:** 10.3390/cancers12113197

**Published:** 2020-10-30

**Authors:** Gaber Plavc, Tanja Jesenko, Miha Oražem, Primož Strojan

**Affiliations:** 1Department of Radiation Oncology, Institute of Oncology Ljubljana, 1000 Ljubljana, Slovenia; morazem@onko-i.si (M.O.); pstrojan@onko-i.si (P.S.); 2Faculty of Medicine, University of Ljubljana, 1000 Ljubljana, Slovenia; tjesenko@onko-i.si; 3Department of Experimental Oncology, Institute of Oncology Ljubljana, 1000 Ljubljana, Slovenia

**Keywords:** head and neck neoplasms, immunoradiotherapy, SBRT, immunotherapy, abscopal effect

## Abstract

**Simple Summary:**

Immunotherapy offers new hope for patients with recurrent or metastatic head and neck cancer. However, only 20% of patients respond to this treatment. Combining radiotherapy in novel ways with immunotherapy can lead to synergistic effect by enabling cancer recognition by immune system and rendering tumor microenvironment less immunosuppressive. Based on a literature review, the main factors that need to be considered in future trials of immunoradiotherapy in head and neck cancer are discussed. The significance of proper timing of the treatment, the radiotherapy fractionation, patient selection, the number and the site of irradiated lesions, and the irradiated volume have been established in preclinical and clinical trials across different solid tumors. However, the trials using immunoradiotherapy in patients with recurrent or metastatic head and neck cancer have shown poor results so far and the reasons for this are elaborated on.

**Abstract:**

Immunotherapy with immune checkpoint inhibitors (ICI) has recently become a standard part of the treatment of recurrent or metastatic head and neck squamous cell carcinoma (R/M HNSCC), although the response rates are low. Numerous preclinical and clinical studies have now illuminated several mechanisms by which radiotherapy (RT) enhances the effect of ICI. From RT-induced immunogenic cancer cell death to its effect on the tumor microenvironment and vasculature, the involved mechanisms are diverse and intertwined. Moreover, the research of these interactions is challenging because of the thin line between immunostimulatory and the immunosuppressive effect of RT. In the era of active research of immunoradiotherapy combinations, the significance of treatment and host-related factors that were previously seen as being less important is being revealed. The impact of dose and fractionation of RT is now well established, whereas selection of the number and location of the lesions to be irradiated in a multi-metastatic setting is something that is only now beginning to be understood. In addition to spatial factors, the timing of irradiation is as equally important and is heavily dependent on the type of ICI used. Interestingly, using smaller-than-conventional RT fields or even partial tumor volume RT could be beneficial in this setting. Among host-related factors, the role of the microbiome on immunotherapy efficacy must not be overlooked nor can we neglect the role of gut irradiation in a combined RT and ICI setting. In this review we elaborate on synergistic mechanisms of immunoradiotherapy combinations, in addition to important factors to consider in future immunoradiotherapy trial designs in R/M HNSCC.

## 1. Introduction

Head and neck cancers account for 3–5% of cancer cases with a 5-year overall survival rate of around 50–65% across all stages in the developed world [1,2,3,4,5]. Two-thirds of these patients present with locally advanced disease in which, despite aggressive multimodal treatment, relapses occur in around 50% of cases in the first two years, whereas 15% of all patients eventually develop distant metastases [6,7]. In patients with recurrent or metastatic (R/M) head and neck squamous cell carcinoma (HNSCC), the most effective systemic treatment offered a median overall survival of 10 months in the pre-immunotherapy era [8]. With the approval of the first immune checkpoint inhibitor targeting immune checkpoint programmed cell death protein 1 (anti-PD-1) for R/M HNSCC in 2016, durable responses are now observed. However, the proportion of responding patients is below 20% and this resistance to anti-PD-1 is usually multifactorial [9,10,11,12]. At the same time, around 60% of HNSCC patients receiving immune checkpoint inhibitors experience immune-related adverse effects and in up to 17% of patients these are grade 3 or higher. However, the overall effect on the quality of life is positive compared to standard chemotherapy treatment [9,10,11,13]. Therefore, research on the possible means of enhancing the anti-PD-1 effect is much needed.

Historically, radiotherapy (RT) was seen as a local treatment and its effect explained by loss of tumor cells’ reproductive ability [14,15]. Furthermore, its net effect on the immune system was understood as immunosuppressive [16]. Now the multifaceted interaction between the immune system and RT is well acknowledged [17]. Even after ablative RT doses the reduction of tumor burden is dependent on functional T cells [18]. Nonetheless, RT serves as an in situ vaccination where it promotes tumor antigen cross-presentation and induces the production of chemokines and cytokines to enhance the local and abscopal antitumor immune responses [19,20].

Despite numerous preclinical studies showing synergistic effects of concomitant anti-PD-1 treatment and RT, the results of clinical trials are not as straightforward [21,22,23]. In this review we present the biological rationale for combining anti-PD-1 with RT in R/M HNSCC and explore the details and formulate recommendations that need to be taken into consideration in design of future clinical trials.

## 2. Biological Rational for Immunoradiotherapy Combination in R/M HNSCC

### 2.1. Resistance to Anti-PD-1 Therapy in R/M HNSCC

Since its discovery in 1991, our understanding of the PD-1/PD-L1 axis has expanded from its role in maintenance of peripheral tolerance to its part in immune evasion of cancer. Binding of the co-inhibitory receptor PD-1 to its ligand PD-L1 results in inhibition of antigen processing and presentation by antigen presenting cells (APC), T cell anergy, and in increase in regulatory T cells. PD-L1 on tumor cells and associated immune cells (combined positive score, CPS) is expressed in 85% patients with R/M HNSCC. Blockade of this axis releases the brake and reinvigorates T cells, resulting in their antitumor activity [10,24,25,26]. Nevertheless, less than one fifth of all patients with R/M HNSCC respond to this therapy and even those who do eventually progress [10,27]. A brief overview of the resistance mechanisms to anti-PD-1 below will be followed by explanation of RT’s potential to modulate them.

A prerequisite for primed CD8+ effector T cells to exert their cytotoxic function is their intratumoral infiltration. Even though HNSCC are among the most immune-infiltrated cancers still less than half of HNSCC are so called inflamed tumors, characterized by ample TILs, inflammatory response, cytolytic activity, and IFN signaling. Furthermore, this immune class of HNSCC can be further dissected into exhausted and active subtypes, with latter having significantly favorable prognosis and showing higher responses to anti-PD-1. These tumors are more likely to be normoxic and be of an inflamed/mesenchymal subtype of human papilloma virus (HPV) mediated tumors [28,29,30,31,32]. The hypoxia is a part of multifaceted immunosuppressive tumor microenvironment, which is defined by the presence of the immunosuppressive metabolites, cytokines, and cells such as regulatory T cells (Tregs), myeloid derived suppressive cells (MDSC), cancer stem-like cells (CSC), and immunosuppressive tumor associated macrophages (TAMs) [33,34,35]. For example, HNSCC has in fact one of the highest Treg/CD8+ T cell ratios among various cancers [32].

After intratumoral infiltration primed CD8+ T cells must recognize specific tumor neoantigens (TNAs, which are not covered by central tolerance) or tumor-associated antigens (TAAs, for which central tolerance is leaky) bound to MHC I on cancer cells [36]. TNAs are a result of mutant peptides mostly resulting from somatic mutations in cancer DNA, and overall tumor mutation burden (TMB) has been shown to correlate with response to anti-PD-1 antibodies across multiple cancers, including HNSCC [37].

Despite HNSCC ranking in the upper quartile of cancers by TMB with five mutations per million base pairs, this by itself is only a prerequisite for effective neoantigen presentation [38,39]. After translation, polypeptides are processed by antigen processing machinery (APM) and loaded onto the MHC I heavy-chain-β2-microglobulin complex. The significance of APM is evident from the lack of CD8+ TIL recognition of HNSCC despite expressed MHC I in the case of defective APM which is deficient in 20–80% of HNSCC [40,41,42]. Loss of β2-microglobulin is also a known evasive mechanism to anti-PD-1 therapy [43,44]. β2-microglobulin gene mutation is uncommon in non-HPV mediated (HPV–) HNSCC in contrast to HPV mediated (HPV+) HNSCC, where genes of the immune presentation pathway, such as β2-microglobulin and HLA, are more often mutated [45,46,47].

Furthermore, inhibition of the PD-1/PD-L1 axis in HNSCC can lead to compensatory upregulation of alternative immune checkpoints, such as TIM-3, LAG-3, CTLA-4, TIGIT, GITR, and VISTA [48,49].

### 2.2. Immunomodulatory Effects of Radiotherapy

The concern for RT’s immunosuppressive effects stems from a generally recognized extreme lymphocytes’ radiosensitivity, resulting in inactivation of 90% of lymphocytes exposed to 3 Gy in in vitro colony formation assay [50]. However, preoperative RT in oral squamous cell carcinoma has been shown to significantly induce proliferative activity of CD8+ TILs and the relative radioresistance of TILs has been attributed to TGFβ which is induced already by low-dose RT [51,52,53,54,55]. Nevertheless, RT can increase the concentration of immunosuppressive cells in HNSCC TME and the magnitude of this effect seems to depend on RT details (e.g., hypofractionated RT increases T cell tumor infiltration, downregulates intratumoral immunosuppressive VEGF, and leads to lesser increase in MDSC as compared to conventionally fractionated RT [56,57,58,59,60,61,62,63]) and on tumor characteristics (e.g., increase of CSCs in TME after RT is more prominent in HPV− HNSCC as compared to HPV+ [64]). Some immunomodulatory effects of RT on TME are presented in Figure 1.

Immunogenic cancer cell death (ICD) leading to the activation of the adaptive response is central to the immunostimulatory effects of various anticancer treatments including RT (Figure 2) [65,66]. In a preclinical model, ICD was shown to be dose-dependent with increasing concentrations of released/expressed danger-associated molecular patterns (DAMPs) that are necessary for the recruitment and maturation of DCs when irradiated from 2 to 20 Gy [65,67,68]. A cross-presentation of antigens by DCs on their major histocompatibility complex I (MHC I) to T cell receptors (TCRs) on naïve CD8+ T cells follows. The latter are activated by co-stimulatory membrane ligands and cytokines provided by DCs upon stimulation by DAMPs and type I interferons (IFN I) [69]. Favorably, RT causes dose-dependent increase of MHC I expression in vitro as well as in vivo [70]. Furthermore, RT also enhances the diversity of TCR repertoire of intratumoral T cells and enhanced diversity of PD-1+CD8+ T cells is a positive predictor of response to anti-PD-1 therapy [71,72]. IFN I production in DCs and cancer cells is driven by exogenous or endogenous DNA in ectopic places such as cytosol which is sensed by a central protein connecting several DNA sensing pathways, Stimulator of interferon genes (STING) [73]. RT produces cytosolic DNA-containing micronuclei and non-mitochondrial free cytosolic DNA, especially in cells with loss of p53 function, which is lost in a majority of HPV– HNSCC [74,75]. Importantly, RT doses above 12–15 Gy induce the production of exonuclease Trex1 that degrades the cytosolic DNA, thus inhibiting type I IFN signaling via cGAS-STING pathway [76]. In HPV+ HNSCC, it was demonstrated that E6 and E7 viral oncoproteins also suppress RIG-I-mediated innate immune signaling and interferon beta (IFNβ) induction [77]. Furthermore, E7 oncoprotein was shown to antagonize the cGAS-STING DNA-sensing pathway and promote degradation of STING, which results in a loss of a IFN I-mediated response [78,79,80]. However, differential responses of HPV+ and HPV− HNSCC to RT in terms of DNA sensing are yet to be elucidated. Notably, sustained IFN I signaling was shown to promote immunosuppression by enhancing PD-L1 and PD-1 expression in HNSCC [81].

To exert their cytotoxic function, primed CD8+ TILs must recognize specific TNAs bound to MHC I on cancer cells [36]. Even though conventional chemotherapy and RT can induce somatic mutations, their contribution to overall TMB is not significant [82,83]. However, it has been shown across several human epithelial cancer cell lines, including HNSCC, that sublethal RT doses have a positive effect on the expression of numerous genes involved in antigen processing and presentation [84].

## 3. Clinical Challenges with Combination of RT and Anti-PD-1 in R/M HNSCC

There are several dozen ongoing trials registered at ClinicalTrials.gov that are exploring the safety and efficacy of ICI combined with RT in R/M HNSCC (Table 1) [85,86,87]. To date, the only randomized trial in metastatic HNSCC utilizing concurrent SBRT and anti-PD-1 therapy to present results is the phase II trial of McBride et al. who randomized 62 patients with metastatic HNSCC to either nivolumab or nivolumab with SBRT (3 × 9 Gy). Only one lesion was irradiated between the first and second dose of nivolumab. There were no significant differences between the two arms in response rates in non-irradiated lesions (34.5% vs. 29.0%), median duration of responses (not reached vs. 9.3 months), median PFS (1.9 vs. 2.6 months), overall survival at one year (50.2% vs. 54.4%), or toxicities grade 3 or higher (13.3% vs. 9.7%) [23]. This same RT regimen, which has been proven to be worth pursuing in previous preclinical and clinical trials, was combined with anti-PD-1 agent cemiplimab in another series of 15 patients with R/M HNSCC who were refractory to one or more lines of prior systemic therapy [76,88,89,90,91]. Three fractions of 9 Gy were delivered one week after the first cemiplimab dose. Low-dose cyclophosphamide and granulocyte-macrophage colony-stimulating factor were added to deplete Tregs and stimulate DC maturation, respectively [91]. With only one partial response this combination failed to show any advantage over anti-PD-1 monotherapy.

There is a plethora of possible reasons for failure of RT to act synergistically with anti-PD-1 in these two trials and these are schematically presented in Figure 3 and further discussed below. As pertinent data in HNSCC are scarce, findings from other tumor types are also used to illustrate the importance of these factors.

### 3.1. Patient Selection

Different patterns of distant metastases (later appearance and diverse sites) have been described in HPV+ versus HPV− oropharyngeal (OP) cancers, which could partially be attributed to survival bias (8-year overall survival of 71% in HPV+ versus 30% in HPV− patients) [92,93,94]. Nevertheless, patients with locoregionally advanced HPV+ OP cancers experience disease recurrence in one third of cases with similar rates of distant metastases as those with HPV− OP cancer [92,93]. HPV infection in these cancers facilitates immune recognition and T cell infiltration. However, higher response to anti-PD-1 therapy in these patients has not been unequivocally confirmed [95,96,97,98]. Due to inherently increased radiosensitivity of HPV+ OP cancers compared to HPV− despite their similar response rates to ICI, locally ablative RT added to ICI could lead to more pronounced synergistic effects by greater reduction of disease burden in HPV+ OP cancer patients [94,99]. On the other hand, in the McBride et al. trial the proportion of responding patients with HPV− HNSCC was higher in the SBRT plus nivolumab arm compared to nivolumab only arm, suggesting that less inflamed HPV− tumors could benefit more from addition of RT [23,95].

Oligometastatic disease is a well-known entity in other cancers, while reports on its occurrence and management in HNSCC are scarce [94]. Nevertheless, aggressive local therapy of (oligo)metastases in HNSCC has been described to offer significant survival advantage [100,101]. Ablative therapy of all accessible metastatic lesions should therefore be strongly considered in the combined immunoradiotherapy setting in patients with oligometastatic HNSCC and good performance status [102,103].

Recurrence in previously irradiated areas is a frequent pattern of relapse in patients with locally advanced HNSCC and it seems that patients with locoregionally recurrent disease as opposed to metastatic disease have somewhat lower responses to anti-PD-1 [10,104,105,106,107]. Furthermore, locoregional recurrence is also a major type of progression on ICI therapy [108]. The question of feasibility and efficacy of re-irradiation during immunotherapy is therefore vital. Two case reports offer some crude answers: a successful re-irradiation of regional oligoprogression during treatment with nivolumab in a heavily pretreated patient with HPV− HNSCC of unknown primary site, and a positive outcome even after a second course of re-irradiation of the recurrence at the primary site in a patient with heavily pretreated metastatic non-keratinizing nasopharyngeal carcinoma receiving pembrolizumab [109,110]. Even though re-irradiation with doses above 50 Gy in combination with chemotherapy by itself offers median overall survival of up to 30% at two years, in both of the presented (re)re-irradiation cases doses were much lower and therefore synergy with immunotherapy is a plausible explanation [107]. These two reports contradict the observed compromised nivolumab efficacy in HNSCC after previous aggressive local treatment reported by some authors; therefore, re-irradiation with its positive effect on TME together with anti-PD-1 could provide valuable treatment intensification in patients with locoregionally recurrent HNSCC [111].

Another important aspect of clinical anti-PD-1 use is the role of concurrent chemotherapy that seems to be appropriate in CPS-low patients with higher symptom burden in need of faster treatment response. Notably, in KEYNOTE-048 duration of response to pembrolizumab alone versus pembrolizumab combined with chemotherapy was substantially longer, 22.6 vs. 6.7 months [10]. It is prudent to speculate that concurrent high-dose chemotherapy with its immunosuppressive effects impairs long-term anti-PD-1 efficacy and that instead of chemotherapy SBRT with its local cytotoxic effect and lesser systemic immunosuppressive effect could be of use for CPS-low patients in need of fast treatment response.

### 3.2. Fractionation and Dose Selection

In general, conflicting results exist regarding the optimal fractionation regimen to elicit a positive immunomodulatory effect of RT, and HNSCC-specific data are lacking.

Experiments on Lewis lung carcinoma, melanoma, colon cancer, and melanoma murine models showed greater potency of SBRT (2 × 11.5 Gy or 3 × 8 Gy) compared to more fractionated RT with comparable dose (9 × 4 Gy or 18 × 2 Gy), resulting in improved local and abscopal antitumor immune response. SBRT superiority was due to greater reduction of intratumoral hypoxia and significantly increased T cell infiltration. When anti-PD-1 was added to fractionated RT, growth suppression was comparable to that of SBRT alone, but inferior to SBRT+anti-PD-1 combination [61,62]. Additionally, in combination with immunotherapy, more fractionated RT regimens resulted in more frequent lymphopenia and inferior outcomes in metastatic lung cancer patients [112].

As described above, ICD is dose-dependent with increasing concentrations of DAMPs when irradiated from 2 to 20 Gy, while doses above 12–15 Gy were shown to inhibit IFN I signaling via cGAS-STING pathway [67,76]. Therefore not surprisingly, delivering 15 Gy in two 7.5 Gy fractions was shown to be more effective in increasing tumor-specific T cell responses without a significant increase in intratumoral Tregs than a single fraction of 15 Gy [113]. Perhaps most well-known is the abscopal effects of RT when combined with anti-CTLA-4 only in fractionated RT (3 × 8 Gy or 5 × 6 Gy) and not in single fraction RT (1 × 20 Gy) in a murine breast cancer model [114]. In a clinical setting, however, Maity et al. observed a clear abscopal response after 1 × 17 Gy used concomitantly with anti-PD-1 in a melanoma patient previously progressing on anti-PD-1 treatment [115].

It is obvious that direct translation of preclinical findings into clinical practice is difficult. Importantly, genetic heterogeneity, TME, tumor vasculature, α/β issue, systemic immune status, and different irradiation techniques in murine models with newly implanted and previously untreated tumors cannot be compared to real-life scenarios [113,116,117]. For example, in immunocompetent mice, the antitumor T lymphocytic response is caused as an artifact of cancer cell implantation and, therefore, tumors treated immediately after implantation are responsive even to single agent immunotherapy. The approach becomes inefficient in the later stages of disease due to the changes in TME [118]. Secondly, contrary to irradiation regimens in preclinical models, in clinical practice SBRT target doses are inherently heterogeneous with maximums of more than 150% of the prescribed dose [119]. Furthermore, in preclinical models, irradiation is performed on superficial tumors, whereas in clinical settings tumors are usually deep-seated and a larger volume of surrounding normal tissue is simultaneously irradiated which could lead to different outcomes. For example, when using conformal multifield RT techniques, a large skin surface is usually irradiated. As a result, highly radioresistant epidermal Langerhans cells upregulate MHC II upon RT, migrate to lymph nodes, and induce an increase in Treg cell numbers, which can be detrimental to the effect of immunotherapy [120]. These factors should be considered when evaluating results from preclinical studies to be translated into clinical trials.

Immuno(radio)therapy is increasingly being tested in the primary setting as either adjuvant, definitive, or neoadjuvant therapy. Findings especially from neoadjuvant trials will provide us with data that can be used in R/M HNSCC as well [121]. For example, in early phase trial preoperative SBRT with either 5 × 8 Gy (*n* = 5) or 3 × 8 Gy (*n* = 5) concurrently with nivolumab, administered 5 weeks pre-surgery in patients with HPV+ OP or unknown primary HNSCC ineligible for transoral robotic surgery yielded 100% complete pathologic response (pCR) in 5 × 8 Gy cohort and 80% pCR in 3 × 8 Gy cohort with major pathologic response (<10% viable cancer cells) in the remaining patient. Toxicity was higher in the 5 × 8 Gy, thus 3 × 8 Gy regimen with comparable clinical effectiveness was selected for further evaluation [89].

Altogether, utilizing hypofractionated RT with moderate doses of 6–12 Gy per fraction seems to be most prudent based on data so far. However, further research on the optimal RT dose and fractionation schedule for its combination with ICI in R/M HNSCC is needed. When the aim of RT delivered concurrently with ICI is control of a single symptomatic site, higher doses offering better local control need to be applied, although at the cost of losing positive immunomodulation.

### 3.3. Site and Number of Lesions

The abscopal effect of RT can be considered clear proof of its in situ vaccination effect but can only be observed if at least one of the tumor lesions is not irradiated [19]. This poses an inherent challenge as disseminated cancers in humans are genetically more heterogeneous than preclinical cancer models and irradiation of a single lesion could therefore be inefficient in serving as an in situ vaccination [116]. This is one of the three major drawbacks of single-site versus multisite irradiation metastatic HNSSC, in which a possibility of a branched pattern of clonal evolution resulting in inter-metastatic heterogeneity is well known [122]. The second is the effect of RT rendering TME less hostile for circulating CD8+ T cells to access and exert their effector function, therefore leaving lesions non-irradiated could be detrimental. Finally, overall tumor burden which cannot be substantially reduced by single-site RT and persisting bulky lesions could present a major obstacle to the effect of immunotherapy. These factors in favor of comprehensive multi-site RT are extensively discussed by Brooks and Chang [123]. An observation by Menon et al. supports multisite irradiation in the setting of immunoradiotherapy even if with only a low dose RT. They analyzed a subset of 26 patients with metastatic cancers (predominantly lung adenocarcinomas) from three prospective trials of immunotherapy with SBRT (mostly 50 Gy in four fractions) which had at least one lesion that received low dose (1–20 Gy, average 7.3 Gy). Low-dose irradiated lesions responded in 58% compared to 18% in no-dose lesions (<1 Gy, *p* = 0.0001). When analyzed per dose ranges, response rate was significantly higher in lesions receiving >5 Gy [124].

Importantly, irradiation of tumor lesions in different organs might lead to different results. McGee at al. showed that irradiation of visceral organs (lung, liver) as opposed to irradiation of brain and bone lesions leads to decreased overall NK cells and their exhaustion. Furthermore, SBRT of viscera was shown to increase and activate memory T cells. Since total dose was associated with these changes and different organs were treated with different SBRT regimens, results of this study must be interpreted with caution [125]. Other researchers showed that the combination of immunotherapy and RT is also effective in brain metastases, however, more treatment-related complications can be anticipated [126,127]. Similar inter-organ differences were observed between liver directed SBRT versus lung directed SBRT with concomitant anti-CTLA-4 agent in a metastatic NSCLC, with a more pronounced increase in T cell activation in the former [128]. Furthermore, when evaluating different immune responses to treatment of visceral versus bone lesions, concomitant therapy with RANKL inhibitors (e.g., denosumab) should be considered an important factor, because anti-RANKL was shown to enhance anti-PD-1 effect in preclinical study [129].

The arguments in favor of multisite SBRT are strong, but caution is advised if ablative doses are used. In a randomized trial of multisite ablative SBRT in patients with various oligometastatic cancers, SBRT-related deaths were observed in 4.5% [130]. Whether irradiation of metastases in different organs in patients with metastatic HNSCC receiving anti-PD-1 leads to different responses is so far unknown.

### 3.4. Timing

Positive effects of RT on immune response are expected early after irradiation. In different murine models RT led to transient increased PD-1 expression on CD8+ and CD4+ T cells 24 h after RT, resulting in synergistic effect of concomitantly administered anti-PD-L1 and RT. This was not a case in a sequential application setting where RT was followed by anti-PD-L1 after 7 days [22]. It seems that application of anti-PD-1 preceding RT could also be of benefit, as was shown in retrospective analysis of 758 patients with different metastatic solid cancers treated with either anti-CTLA-4 or anti-PD-1/L1, and RT within 30 days of immunotherapy. Those patients receiving immunotherapy 30 days or more before RT benefited the most regardless of histology, immunotherapy type, and anatomic site of RT [131]. This could, however, be a consequence of the fact that early progressors who usually have worse prognosis were being treated by RT earlier. Importantly, half-lives of anti-PD-1 agents are rather long, with serum half-life ranging from 12 to 23 days with peak occupancy of circulating CD3 T cells 4–24 h and a plateau occupancy 57 days after application [132,133].

Another point in favor of concomitant RT and anti-PD-1 treatment is RT’s local cytotoxic effect. Hyperprogression during ICI therapy has been described in 0–29% of patients with R/M HNSCC and appears to be more common in younger patients, in those with regional recurrences, and/or higher peripheral neutrophil–lymphocyte ratio [134,135,136]. Adding RT early during immunotherapy could potentially prevent hyperprogression and its detrimental effect.

An often overlooked aspect of timing is inter-fractional time interval of SBRT, which deserves special attention in hypoxic solid tumors such as HNSCC [137]. Hypoxia makes tumors more resistant to radio- and immunotherapy [137,138]. While SBRT is known to induce considerable vascular damage resulting in postponed indirect cell death, normalization of tumor vasculature after ablative SBRT has also been observed. These differences in the observed outcomes are probably a consequence of different RT fractionation and dose regimens, and different TMEs [139,140]. Nevertheless, prolonging inter-fractional interval results in reoxygenation (largely due to reduced oxygen consumption that follows extensive cancer cell death), thus removing hypoxia-induced immunosuppressive stimuli and radioresistance, which can be an issue particularly in bulky tumors [141].

Research on the optimal timing of combining immunotherapy and RT optimal timing should also take into consideration the circadian rhythms of different parts of the immune system. Intrinsic clocks affect innate as well as adaptive immune cells’ development, movement, and function. For example, egress of T cells into efferent lymphatics in a preclinical murine model was observed to be highest around 9 h after dawn and vaccination of the elderly was shown to be more effective in inducing antibodies production when done in the morning as opposed to the afternoon [142]. In addition to circadian oscillations, immune oscillations over several days were also observed to have an impact on the efficacy of cancer therapy and should be further explored [143,144,145].

Even though HNSCC-specific data on optimal timing of RT and anti-PD-1 are lacking, some conclusions can be made from experiences in preclinical setting and in other cancers. RT should precede anti-PD-1 or be applied concomitantly, while prolongation of the inter-fractional time interval could be beneficial in bulky/hypoxic tumors. In addition, further fine-tuning of the timing in relation to immune circadian rhythms and daily oscillations is warranted.

### 3.5. Field Selection and Dose Heterogeneity

Elective nodal irradiation targets possible subclinical microscopic disease in draining lymph nodes (DLN). Risk-adapted electively irradiated volume and reduction of dose are areas of active research for reasons including the role of DLN in generating tumor-specific effector CD8+ lymphocytes which may be hindered by RT [146]. For example, surgically DLN-ablated or genetically DLN-deficient mice are found to exhibit a marked decrease in local tumor RT efficiency due to a significant reduction of tumor-specific CD8+ TILs [147]. In immunoradiotherapy, combination care should be taken to spare DLN of, ideally, any irradiation.

To avoid marginal misses, careful tumor delineation with an additional margin to ensure an adequate dose coverage of the target with a predefined probability is a standard in modern RT. However, results from preclinical studies show that partial tumor irradiation of 50% of tumor volume is non-inferior to that of full-volume irradiation with the same dose. In the non-irradiated half an increase in endothelial cells’ adhesion molecules expression as well as three-fold increase in CD8+ T cell concentration, originating from the irradiated half or tumor periphery, were observed. Hemi-irradiation also elicited abscopal effect which was of comparable magnitude to that after whole tumor irradiation. Notably, using a higher dose of 20 Gy had no hemi-irradiation effect. Authors speculated that vascular damage by 1 × 20 Gy prevented immune cell infiltration [148]. In fact, this higher dose could result in detrimental induction of exonuclease Trex1 [76]. The clinical experience seems to concur with these results. Seventy-nine patients with metastatic cancers, of which four had HNSCC, received SBRT in various fractionations to 2–4 metastases which was followed by pembrolizumab within 7 days after SBRT. Metastases larger than 65 cm^3^ were irradiated only partially. Nevertheless, at 6 months there was no difference in local control between fully and partially irradiated lesions [149,150]. Altogether this is a strong argument in favor of partial tumor irradiation when full-lesion coverage would lead to unnecessary DLN or normal peripheral tissue irradiation.

Although a less potent inductor of immunogenic cell death, low-dose RT (0.5–2 Gy) has the advantage of activating antitumoral activity of the TME by increasing the ratio between classically activated type 1 antitumoral macrophages versus alternatively-activated type 2 tumor promoting macrophages in TME, and inducing the expression of cell adhesion molecules such as ICAM-1 or E-selectin [151,152]. It was shown in a murine model that ablative irradiation with 22 Gy in a single fraction followed by primary tumor low-dose RT of 4 × 0.5 Gy results in improved survival [153]. Interestingly, whole-lung low-dose RT that followed ablative RT to the primary tumor also prolonged survival by decreasing immunosuppressive cell concentrations in this metastasis-prone organ. The same effect with increased effector CD8+ T cells and decreased Tregs was observed after delivering low-dose RT to metastatic sites that followed ablative RT of the primary tumor [153]. This are all points in favor of (multisite) SBRT where extensive volume of normal tissue receives low doses of RT. Furthermore, if above described high-dose partial tumor irradiation is used, tumor periphery receives low doses which could be beneficial to lymphocytes adhesion and extravasation.

When re-irradiation is indicated in HNSCC patients, careful consideration of radiation tolerance of surrounding normal tissue is mandatory. For example, one of the most dreadful complications is carotid blowout syndrome which occurs in up to 16% of reirradiated patients if every day SBRT is used [154]. Besides patient selection, careful sparing of peritumoral tissue by only partially irradiating these recurrences could provide above described benefits to concurrent immunotherapy without causing severe toxicity. Researchers from Klagenfurt reported on 23 patients with different bulky tumors that received SBRT (1–3 fractions of 10–12 Gy to 70% isodose line; 65% received 1 × 10 Gy) only to the hypoxic segment inside the tumor volume (defined by positron emission tomography and contrast-enhanced computed tomography as the hypovascularized and hypometabolic segment). Peritumoral tissue and DLNs were defined as the organ at risk and received as low a dose as possible. In this retrospective study they observed response rates (30% or greater regression) in 96% of irradiated lesions and abscopal responses in 52% of patients [90]. Contrary to all the mechanisms of radiation induced antitumor immune response described above, a surprising response was observed in a patient with synchronous squamous cell carcinoma and adenocarcinoma of the lung. This patient neoadjuvantly received 3 × 10 Gy to a hypoxic subvolume of initially unresectable lung squamous cell carcinoma (with metastatic mediastinal lymph nodes and a separate lung adenocarcinoma both outside of the treatment field). Both primaries and DLN were then surgically removed and histological evaluation showed a complete response in DLN and 80% necrosis in nonirradiated separate lung adenocarcinoma, which was not accompanied by substantial TILs. However, apoptosis-inducing factor was highly upregulated [90]. Overall, such a high rate of abscopal responses (i.e., 52%) has not been reported to date in irradiated-only patients and an abscopal response has never been described in a metastatic lesion with a histology differing from that of the irradiated one. Hopefully, these results will be confirmed in a prospective trial. This study poses many questions and results are not in line with our current understanding of the RT induced abscopal response.

Importantly, this peritumoral tissue-sparing technique also delivers a lower dose to the invasive tumor border, where tertiary lymphoid structures (TLS) are often found [155]. TLS are important for initiating and maintaining antitumor immune response and were also shown to be a favorable prognostic factor in HNSCC [155]. The presence of TLS was found to be associated with increased tumor apoptosis and increased radiosensitivity of CD3 cells in TLS compared to other intratumoral CD3 cells has been observed [156]. Mature DC cells defined by expression of LAMP (lysosomal associated membrane glycoprotein) are also detected almost exclusively in the peritumoral areas [157]. Therefore, relative sparing of the invasive tumor border from a high RT dose could also preserve their function.

### 3.6. Other Outcome Defining Factors and Evaluation of Immunoradiotherapy Efficacy

There are some other factors that need to be considered when evaluating the responses to immunoradiotherapy. For example, previous (chemo)radiotherapy for primary HNSCC was shown to confer a better prognosis in patients receiving ICI in R/M setting [158,159]. Reasons for this are unknown and can only be speculated on in view of the findings from preclinical studies, including upregulation of PD-L1 by previous RT. The predictive role of combined positive score (CPS, the number of PD-L1 positive cancer cells, lymphocytes, and macrophages, divided by the number of viable cancer cells and multiplied by 100) was supported by the results from the KEYNOTE-048 trial. In patients with CPS ≥ 1 pembrolizumab monotherapy resulted in significantly superior overall survival compared to standard of care systemic treatment [10]. In single arm trials of immunoradiotherapy combination in HNSCC this factor should therefore be included in the analyses.

Another factor that needs to be considered when seeking to explain the differences in outcomes of immunoradiotherapy in HNSCC patients is microbiome. After establishing a connection between gut microbiome and immune checkpoint inhibitor efficacy in preclinical murine models in 2015, the essential role of microbiome was later confirmed in epithelial cancer patients [160]. Specifically, an abundance of *Akkermansia muciniphila* in feces was associated with better responses and prognosis in these patients treated with anti-PD-1, whereas antibiotic use was an independent negative predictive factor for its efficacy [160,161]. This is rather unfortunate as patients with HNSCC are at an increased risk of soft tissue and respiratory tract infections that often need to be treated with broad-spectrum antimicrobial agents. Indeed, a recent series of 272 patients with locally advanced HNSCC reported that nearly half (45.6%) received antibiotic treatment and these patients had significantly shorter progression-free and disease-specific survival [162]. However, all antibiotics are probably not equally detrimental to the effect of immunotherapy. Recently a positive effect of peroral vancomycin, a Gram-positive-bacteria targeting antibiotic which is retained within the gut, on antitumor immune response and tumor growth was reported in irradiated melanoma and HPV E6/7 expressing lung and cervical cancer murine model. Vancomycin depleted butyrate producing gut bacteria which resulted in reduced concentration of butyrate in fecal and tissue samples. As butyrate otherwise impairs DC antigen presentation to CD8+ T cells, vancomycin enhanced direct and abscopal antitumor activity of hypofractionated RT [163].

Regarding the strong influence of intestinal microbiota on the effect of anti-PD-1 treatment, little is known about the influence of irradiation of intestines to gut microbiome disturbance and, consequently, on antitumor immunity [164,165]. For example, 4–5 Gy of gut irradiation results in a marked increase of commensal Bifidobacterium in intestinal flora and compromises the integrity of the gut leading to translocation of *Enterobacter cloacae*, *Escherichia coli*, and Bifidobacterium into mesenteric lymph nodes and of lipopolysaccharide into serum. This results in an increased DCs and enhanced CD8+ T cell priming by DCs as well as in increased concentrations of systemic inflammatory cytokines resulting in enhanced effect of immunotherapy [166,167,168]. This could potentially even have a role in differences in T cell activation between liver and lung directed SBRT described by Tang et al. (see above), because in liver RT at least part of the gut is irradiated [128]. As seen here, gut irradiation can have a profound impact on the gut microbiome and its effect on immune response. Therefore, irradiation of intestines should also be considered as an outcome-influencing factor in future trials.

While we await further clinical data on the interplay between gut microbiome, ICI, and RT, antibiotics should be used prudently and as targeted as possible.

Some possible pitfalls in ascribing synergistic interaction between RT and ICI are apparent from a case report of a patient with metastatic HPV+ HNSCC who progressed during treatment with a combination of nivolumab and ipilimumab. At this point palliative RT was delivered to the neck while the patient continued with immunotherapy. After two weeks the lung metastasis decreased by 50% despite not being irradiated. Even though pseudoprogression was observed in only 2–4% of HNSCC cases treated with ICI, this possibility should be kept in mind when progression to ICI is suspected [169,170]. The same goes for late response to anti-PD-1 treatment, as the median time to response is around 1.5–9.1 months [9,10,97]. It is therefore uncertain if this was indeed an abscopal response of RT.

Numerous trials testing various combinations of RT and immune checkpoint inhibitors with or without additional agents are still underway [17,85,171]. Interestingly, the addition of one or two applications of low-dose cisplatin, a backbone of the standard HNSCC chemoradiation regimen, was also shown to strongly enhance the abscopal effect when used in combination with RT and anti-PD-1 in murine melanoma, adenocarcinoma, breast cancer, and colon cancer models [172,173]. Major parameters to be considered in planning immunoradiotherapy trials in R/M HNSCC are summarized in Table 2.

## 4. Conclusions

A growing body of evidence has established the potential of RT to act synergistically with immunotherapy in various cancers. In the light of disappointing response rates to anti-PD-1 alone in R/M HNSCC, active research is being conducted on combinations with novel drugs and, in particular, RT. To date, the results of clinical trials exploring combinations of RT with ICI in HNSCC are not consistent. Vital questions regarding patient selection for this treatment combination, e.g., impact of HPV status, recurrent versus metastatic disease, CPS-low versus CPS-high, are worthy of further research. It is safe to say that targeting single lesions in metastatic HNSCC does not enhance the anti-PD1-1 effect and therefore ideally all the metastatic lesions should be targeted, while special attention must be paid to sparing peritumoral and nontumoral tissue even if at the expense of full lesion volume coverage. Resulting intratumoral dose heterogeneity could be beneficial. Based on preclinical and clinical data from other cancers, RT should be delivered concurrently with or immediately before anti-PD-1, employing small conformal fields of SBRT and multiple intermediate-dose fractions (6–15 Gy).

However, the details leading to maximum synergism between RT and ICI in R/M HNSCC are far from being well established and the present knowledge offers only a glimpse into the multitude of factors needed to be considered in future immunoradiotherapy trial designs.

## Figures and Tables

**Figure 1 cancers-12-03197-f001:**
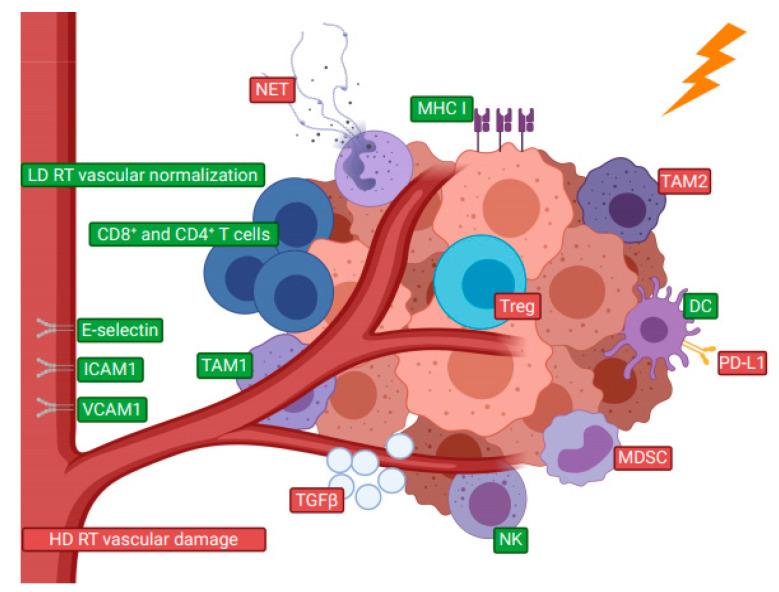
Selected immunomodulatory effects of radiotherapy (RT) on tumor microenvironment (TME). Depending on the dose and fractionation as well as characteristics of established TME, RT can act immunostimulatory by enhancing major histocompatibility complex I (MHC I) expression, antitumoral type 1 tumor-associated macrophages (TAM1), natural killer cells (NK), dendritic cells (DC) and cytotoxic (CD8+) and helper (CD4+) T cell infiltration. Contrary, the RT impact can be immunosuppressive by inducing protumoral type 2 tumor-associated macrophages (TAM2), regulatory T cell (Treg) and myeloid-derived suppressive cell (MDSC) infiltration, programmed-death ligand 1 (PD-L1) upregulation, tumor growth factor β (TGFβ) expression, and neutrophil extracellular traps (NET) expulsion. The impact of low dose RT (LD RT) and high dose RT (HD RT) on tumor vasculature is differential. Furthermore, RT induces adhesion molecules expression on endothelium, such as intercellular adhesion molecule 1 (ICAM-1), vascular cell adhesion molecule 1 (VCAM-1), and E-selectin, which are needed for leukocytes to extravasate and infiltrate tumors. (Created with BioRender.com).

**Figure 2 cancers-12-03197-f002:**
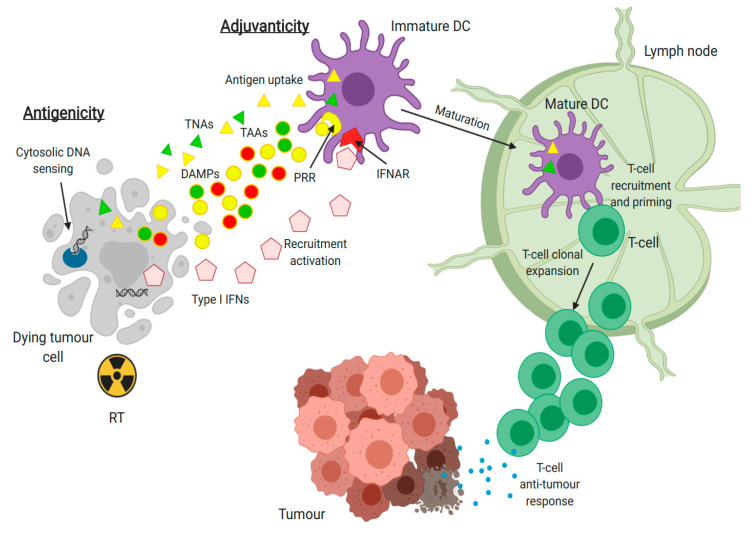
Simplified presentation of radiation-induced immunogenic cell death. Damage induced by RT mediates the release of tumor neoantigens (TNAs) or tumor-associated antigens (TAAs) and potentiates the antigenicity of the cancer cells. The induced antigenicity of tumors must be coordinated by the exposure of danger signals (DAMPs) that are necessary for the recruitment and maturation of antigen-presenting cells such as dendritic cell (DC). This is referred to as adjuvanticity. Therefore, dying cells must expose or release DAMPs which are recognized by pattern recognition receptors (PRR) on DCs. This results in recruitment and enhanced uptake of cancer antigens by DCs. Type I IFNs are sensed by interferon-alpha/beta receptor (IFNAR) in DCs, which is vital for antigen cross-presentation to CD8+ T cells and their activation, resulting in antitumor response. (Created with BioRender.com).

**Figure 3 cancers-12-03197-f003:**
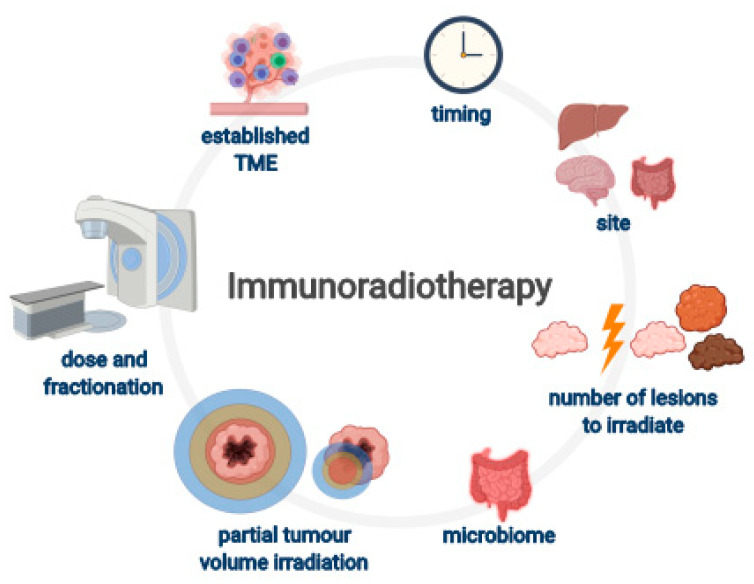
Factors to consider in trials of combined radiotherapy (RT) and immunotherapy. Even though the most effective RT dose-fractionation in the setting of immunoradiotherapy is still ill-defined, biological rationales for particular RT schemes are being elucidated. Due to inherent radiosensitivity of lymphocytes and different mechanisms of actions of immunotherapeutics, temporal coordination of RT and immunotherapy is most important. Reducing overall tumor burden and interlesional heterogeneity point in favor of multisite irradiation vs. single-site irradiation in metastatic setting. It also seems irradiation of lesions in different organs, e.g., in liver vs. in brain, leads to different immunogenic results. Importantly, partial tumor volume irradiation appears to confer compelling results. The characteristics of the existent immunosuppressive tumor microenvironment (TME) are heterogeneous and dynamic and present a key obstacle to (radio) immunotherapy efficacy. (Created with BioRender.com).

**Table 1 cancers-12-03197-t001:** Ongoing clinical trials incorporating radiotherapy and immunotherapy in recurrent or metastatic head and neck squamous cell carcinoma.

Name	Study Phase	(Planned) Number of Participants	Immunotherapy	RT fx	RT Technique	RT Target	Timing of RT and ICI	Primary Endpoint
CONFRONT, NCT03844763	I–II	71	Avelumab and cyclophosphamide	1 × 8 Gy	“Highly conformal”	1 lesion	ICI 1. day and Q2W; RT 8. day	Toxicity and ORR
NCT03283605	I–II	35	Durvalumab and tremelimumab	30–50 Gy in 3–5 fx	SBRT	2–5 lesions	RT between 2. and 3. ICI cycle	PFS
NCT03539198	I	91	Nivolumab	3–5 fx (various doses)	Proton SBRT	1 lesion	RT between 2. and 3. ICI cycle	ORR
REPORT, NCT03317327	I–II	20	Nivolumab	60 Gy in 1.5 Gy fx BID	NA	LR recurrence or 2. primary	RT starts with the 2. ICI cycle	Toxicity
NCT03522584	I–II	20	Durvalumab and tremelimumab	3 fx (dose unknown)	HIGRT or SBRT	1–5 lesions	RT during week 3 of ICI	Toxicity
NCT02684253	II, randomized	65	Nivolumab	3 × 9 Gy (randomized to nivolumab +/− RT)	SBRT	1 lesion	RT between 1. and 2. cycle of ICI	ORR
NCT03521570	II	51	Nivolumab	Unknown (completed in 6–6.5 weeks)	IMRT	LR recurrence or 2. primary	RT starts with the 2. ICI cycle	PFS
NCT02289209	II	48	Pembrolizumab	60 Gy in 1.2 Gy fx BID	NA	LR recurrence or 2. primary	RT starts with the 1. ICI cycle	PFS
NCT03085719	II	26	Pembrolizumab	High dose in 3 fx and low dose in 2 fx	NA	Minimum 1 lesion	NA	ORR
KEYSTROKE, RTOG 3507, NCT03546582	II, randomized	102	Pembrolizumab	NA (over 2 weeks; randomized to SBRT+/− pembrolizumab)	SBRT	LR recurrence or 2. primary	SBRT and then ICI	PFS
Keynote-717, IMPORTANCE, NCT03386357	II, randomized	130	Pembrolizumab	12 × 3 Gy (randomized to pembrolizumab +/− RT)	NA	1–3 lesions	ICI on the 3. day of RT	ORR
NCT04454489	II	15	Pembrolizumab	Quad-shot RT	NA	At least 1 lesion in the head and neck region	RT starts between ICI cycles 2 and 3	ORR
NCT04399785	II	34	Camrelizumab	NA	SBRT	NA	NA	ORR

RT—radiotherapy, fx—fractions of RT, ICI—immune checkpoint inhibitor, Q2W—every other week, ORR—overall response rate, SBRT—stereotactic body RT, PFS—progression-free survival, BID—two fraction in a day, NA—not available, LR—locoregional, HIGRT—hypofractionated image-guided RT, IMRT—intensity-modulated RT, Quad-shot—at least one cycle of 14.8 Gy in four fractions (3.7 Gy per fraction) delivered twice daily over two consecutive days.

**Table 2 cancers-12-03197-t002:** Radiotherapy-related parameters to be considered in future clinical trials on combination of radiotherapy and immune checkpoint inhibitors in patients with recurrent or metastatic squamous cell carcinoma of the head and neck.

Parameter	Recommendation	Explanation
RT regimen	SBRT (multiple fractions of 6–15 Gy)	- Suppressing MDSC recruitment [62] - Increasing TILs [61] - Favorable tumor-reactive T cells/Treg ratio [113] - Minimizing RT-induced systemic lymphopenia [112,174]
Number of lesions	Majority of lesions	- Targeting genetically heterogeneous metastatic lesions [116] - Reducing overall tumor burden [123] - Rendering TME more favorable to TILs infiltration [123]
Timing of RT	Concurrent or close to ICI	- Concurrently or right before anti-PD-1 [22] - Could prevent detrimental hyperprogression in anti-PD-1/L1 treated R/M HNSCC patients [134]
Selection of RT field	Tumor-only	- Omitting DLN irradiation is beneficial [175]
Dose heterogeneity	Consider delivering high dose to partial tumor volume	- Could lead to similar results as full-tumor-volume-RT [148,149,150] - Beneficial effects of low-dose RT [153] - Endothelial cell adhesion molecules induced by 1–5 Gy [152] - Relative sparing of tumor invasive border and peritumoral tissue could be beneficial [90,155,157]

RT—radiotherapy, SBRT—stereotactic body RT, MDSC—myeloid-derived suppressive cells, TILs—tumor infiltrating T lymphocytes, Treg—regulatory T cells, TME—tumor microenvironment, ICI—immune checkpoint inhibitors, PD-1—programmed cell death protein-1, CTLA-4—cytotoxic T-lymphocyte-associated protein 4, PD-L1—programmed cell death protein-ligand 1, R/M HNSCC—recurrent/metastatic head and neck squamous cell carcinoma, DLN—draining lymph nodes.

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
