# Peer review of "Challenges in Combining Immunotherapy with Radiotherapy in Recurrent/Metastatic Head and Neck Cancer"

_cancers, 2020, doi:10.3390/cancers12113197_

Round 1

Reviewer 1 Report

The authors have thoroughly addressed all comments and suggestions.  The review is well written and much easier to follow. This is an important contribution to the field and nicely reviews the RT and immunotherapy including challenges, benefits and future research required. I recommend acceptance.

Reviewer 2 Report

No further comments, the flow of the manuscript has been much improved.

This manuscript is a resubmission of an earlier submission. The following is a list of the peer review reports and author responses from that submission.

Round 1

Reviewer 1 Report

The authors report on an extensive review aimed to unravel the biologic background and rationale in support of the combination of radiotherapy and immune checkpoint inhibitors in recurrent/metastatic head and neck cancer. The authors should be commended for having produced a very complete and up-to-date manuscript. In particular, the biologic interplay which lays the ground for the interplay between radiotherapy and the immune system is very deeply described.

I have however a major concern regarding the structure of the text and its flow. In the actual form, in my opinion the text is not in line with what the title of the article would hint at. In particular, a large, predominant part of the text is devoted to a very focused description of well described biological mechanisims, which however in most cases are not specific of head and neck cancer only. Rather, these sections (from 2.1 to 2.6 and similarly from 4.1 to 4.5) refer to detailed in vitro and early clinical experiences which are however in most cases independent from a specific tumor type. By reading “Challenges in combining immunotherapy with radiotherapy in recurrent/metastatic head and neck cancer” one would expect a much different content. The reader could come up with some relevant – yet unaddressed questions – such as

  • what is the predominant pattern of failure of immunotherapy in HNSCC? Are there data to suggest an influence of previous radiotherapy (fields, dose, fractionation, time to immunotherapy, etc)?
  • are there any relevant differences between the data regarding Nivolumab in second line/platinum refractory setting vs Pembrolizumab (plus/minus CT) in first line?
  • are there any predictive factors of different clinical behavior (i.e, HPV status? CPS?)
  • what is the actual rate of hyperprogression? are there any data that can correlate this phenomenon with previous RT fields?
  • are there any data supporting the role of RT (SBRT) in the context of oligometastatic HNSCC undergoing immunotherapy?
  • are there any data in regards to oligoprogression in the context of immunotherapy for HNSCC? Potential of RT in this field?
  • can the authors condense biological aspects as speculations to justify a paradigm shift in the design of rt fields in the primary setting (such as avoiding N stations in the elective neck to reduce the risk of regional lymphocyte depletion?) or hypothesize a correlation between rt fields and pattern of failure which could be somehow recapitulated in the recurrent/metastatic setting under immunotherapy?

Overall, the text is well written and very informative but in its actual from is not reader-friendly and far too long. It is also pretty disorganized in terms of not corresponding in a single train of thought to what the title of the paper would allude to.

If the authors agree, I would suggest

  • to largely reduce the text pertaining to the biological background on the interplay between rt and immune system, trying to condense the presented data in a more focused way on HNSCC
  • to provide the text with a a more focused clinical orientation, which could be highly interesting to speculate on. The biological data may then be used to provide the reader with some context and serve as a scientific background to all the relevant clinical questions that could be addressed, as for instance suggested above.
  • to provide the reader with some take-home messages on how radiotherapy could be used to enhance the efficacy of immune check point inhibitors in the R/M setting

As an alternative, the title of the paper and its content should be refocused and presented as an exclusive “biological” review rather than a blend between many preclinical/early clinical data and a very specific clinical question pertaining to head and neck cancer.

Author Response

The authors would like to thank both reviewers for in-depth comments. Besides other specific recommendations, both reviewers recommended shortening of the first part of the manuscript to put emphasis on the clinical aspect and to make the text more coherent.

We have shortened the first part of the manuscript (biological rational for immunoradiotherapy combination) and changed the narrative in such a way that biological background now supports and explains clinical evidence and clinical recommendations for recurrent or metastatic head and neck cancer.

Other corrections are described in the attached file and were recorded in the manuscript by tracking changes.

Reviewer 2 Report

Overall this is a well written and comprehensive review of the effects of radiation and the tumor response and potential mechanisms of resistance and ways to potentially mitigate the latter in combination with immunotherapy.

Moderate specific points:

Ref 1 is a bit outdated- would recommend more recent reference

PDL1 has been shown be to upregulated on more than colon and mammory cells in response to RT. would include more references for other tumor types such as HNSCC and lung. (Sect 2.3)

Sect 2.4:It is still controversial on whether anti-PD1 results in better response in HPV+ tumors

Would avoid questions as subtitles (Sect 2.5)

In section 3, there is a long explanation of singular case reports- this seems odd in a review and would be better to discuss multiple case reports in combination or larger studies that have demonstrated abscopal effect to condense some of this section. What should the reader get out of your review on this topic?

It would be good to give a review of differences and similarities between tumor types rather than individually discussing in each section. 

Microbiome is mentioned in two different areas- would recommend combining and expanding this area a bit. 

Overall, condensing some of the information would be ideal. Although comprehensive, there is a lot of information making it a bit difficult to pull out the important points. Most sections are recaps of articles rather than a review of what it all means. Needs to be easier on reader to come away and understand where we are with radiation and immunotherapy and what are the questions that still need answered.  Some rearranging and emphasis would help.

Minor:

Some very minor formatting issues (subtitles with no space)

Section 2.4 HNSCC is introduced but I don't think it had been spelled out yet in manuscript. 

Author Response

The authors would like to thank both reviewers for in-depth comments. Besides other specific recommendations, both reviewers recommended shortening of the first part of the manuscript to put emphasis on the clinical aspect and to make the text more coherent.

We have shortened the first part of the manuscript (biological rational for radioimmunotherapy combination) and changed the narrative in such a way that biological background now supports and explains clinical evidence and clinical recommendations for recurrent head and neck cancer.

Other corrections are described in the attached file and were recorded in the manuscript by tracking changes.

Round 2

Reviewer 1 Report

I went trough the revised version of the manuscript carefully.
The authors have to be commended for their efforts in producing such a comprehensive review.
However, I'm afraid that my main concern has not been adequately taken care of.
The paper is very informative but  - in my opinion - is not reader-friendly: from my point of view, the train of thought is not clear. Is the reader supposed to follow all the (very precisely reported) biological assumptions on the interplay between RT and immunotherapy and build on them to figure out which are the clinical challenges in the recurrent/metastatic setting for head and neck cancer? As already shared in my first comments to the paper, I'm personally a bit puzzled by what is the main message that the authors would like to convey. Is this a comprehensive biological review on a very interesting topic - immuno + RT - or a paper more focused towards a clinical question (recurrent/metastatic HN cancer) which one would expect to be covered more in depth? The content of the paper is not clearly focused on either of them - at least not on the clinical argument. The conclusions provided by the authors reflect the imbalance in the previous sections.
 I'm afraid my main concerns have not been solved, as you can see from my initial comments and the authors' reply in their cover letter.

Author Response

We carefully revised the manuscript according to reviewers' comments and major editing of the text was performed.

As suggested by the reviewer, the extensive details of the biological background were removed, and the storyline of the review is now more straightforward and more akin to the natural flow of the reader’s thought process. First, the biological rational for immunoradiotherapy combination in recurrent/metastatic head and neck squamous cell carcinoma (R/M HNSCC) is presented. The problem of resistance to anti-PD-1 in R/M HNSCC is briefly described, which is followed by a short biological background on immunomodulatory effects of radiotherapy (RT). This biological background has been further shortened and is now even more HNSCC-oriented. The text is now more clinically oriented, and emphasis is now on the main part of the article in which we address the clinical nuances pertinent to successful anti-PD-1 and RT combination (immunoradiotherapy) in R/M HNSCC. Due to the paucity of HNSCC-specific data, pertinent results from other research in other histologies were occasionally used in the manuscript. Biological data are used to provide the scientific background to the clinical observations. Beside critically reviewing current available data, relevant questions that still need to be answered are highlighted. Furthermore, take-home massages are presented as specific recommendations regarding major parameters to be considered when planning immunoradiotherapy trials in R/M HNSCC.

We believe this manuscript offers concise overview of the current immunoradiotherapy field in R/M HNSCC, which is full of unknowns, and underscores questions that demand further research. We hope that after this extensive editing, the paper is in accordance with the reviewers’ comments and is now suitable for publication.